# Systematic, comprehensive, evidence-based approach to identify neuroprotective interventions for motor neuron disease: using systematic reviews to inform expert consensus

Charis Wong [1,2,3,4] Jenna M Gregory,[2,3,5] Jing Liao,[3] Kieren Egan [3,6] Hanna M Vesterinen,[3] Aimal Ahmad Khan,[7] Maarij Anwar [7] Caitlin Beagan,[7] Fraser S Brown,[1,3] John Cafferkey,[8] Alessandra Cardinali,[2,3,9] Jane Yi Chiam,[7] Claire Chiang,[7] Victoria Collins [7] Joyce Dormido,[10] Elizabeth Elliott,[1,2,3] Peter Foley,[1,3] Yu Cheng Foo,[7] Lily Fulton-Humble,[5] Angus B Gane,[11] Stella A Glasmacher [1,3] Áine Heffernan [9] Kiran Jayaprakash,[1,3] Nimesh Jayasuriya,[8,11] Amina Kaddouri,[7] Jamie Kiernan,[7] Gavin Langlands,[12] D Leighton,[2,3,13] Jiaming Liu,[7] James Lyon,[8] Arpan R Mehta,[1,2,3,9] Alyssa Meng,[14] Vivienne Nguyen,[7] Na Hyun Park,[8] Suzanne Quigley,[1,3] Yousuf Rashid,[7] Andrea Salzinger,[3,9] Bethany Shiell,[11] Ankur Singh,[11] Tim Soane,[15] Alexandra Thompson,[11] Olaf Tomala [7] Fergal M Waldron,[5,16] Bhuvaneish T Selvaraj,[1,2,3,9] Jeremy Chataway [4,17,18] Robert Swingler,[2] Peter Connick,[1,3] Suvankar Pal [1,2,3] Siddharthan Chandran [1,2,3,9] Malcolm Macleod [3]

► http://dx.doi.org/10.1136/bmjopen-2022-064173

**Correspondence to**
Malcolm Macleod;
Malcolm.macleod@ed.ac.uk

## ABSTRACT

**Objectives** Motor neuron disease (MND) is an incurable progressive neurodegenerative disease with limited treatment options. There is a pressing need for innovation in identifying therapies to take to clinical trial. Here, we detail a systematic and structured evidence-based approach to inform consensus decision making to select the first two drugs for evaluation in Motor Neuron Disease-Systematic Multi-arm Adaptive Randomised Trial (MND-SMART: NCT04302870), an adaptive platform trial. We aim to identify and prioritise candidate drugs which have the best available evidence for efficacy, acceptable safety profiles and are feasible for evaluation within the trial protocol.

**Methods** We conducted a two-stage systematic review to identify potential neuroprotective interventions. First, we reviewed clinical studies in MND, Alzheimer's disease, Huntington's disease, Parkinson's disease and multiple sclerosis, identifying drugs described in at least one MND publication or publications in two or more other diseases. We scored and ranked drugs using a metric evaluating safety, efficacy, study size and study quality. In stage two, we reviewed efficacy of drugs in MND animal models, multicellular eukaryotic models and human induced pluripotent stem cell (iPSC) studies. An expert panel reviewed candidate drugs over two shortlisting rounds and a final selection round, considering the systematic review findings, late breaking evidence, mechanistic plausibility, safety, tolerability and feasibility of evaluation in MND-SMART.

## STRENGTHS AND LIMITATIONS OF THIS STUDY

⇒ We describe a systematic, structured and evidence-based, consensus approach for drug repurposing in motor neuron disease (MND), specifically for Motor Neuron Disease – Systematic Multi-arm Adaptive Randomised Trial, a phase III multi-arm multi-stage adaptive clinical trial in MND.

⇒ Systematic reviews of clinical studies in neurodegenerative diseases and MND preclinical studies provide a robust evidence base to inform expert panel decisions on drug selection for clinical trial.

⇒ Providing a contemporary evidence base using traditional systematic reviews is challenging given their time-consuming and labour-intensive nature.

⇒ Incorporation of machine learning and automation tools for systematic reviews, and data from experimental drug screening can be helpful for future drug selection.

**Results** From the clinical review, we identified 595 interventions. 66 drugs met our drug/disease logic. Of these, 22 drugs with supportive clinical and preclinical evidence were shortlisted at round 1. Seven drugs proceeded to round 2. The panel reached a consensus to evaluate memantine and trazodone as the first two arms of MND-SMART.

**Discussion** For future drug selection, we will incorporate automation tools, text-mining and machine learning

techniques to the systematic reviews and consider data generated from other domains, including high-throughput phenotypic screening of human iPSCs.

## INTRODUCTION

Motor neuron disease (MND), also known as amyotrophic lateral sclerosis (ALS), is a progressive neurodegenerative disease with a median survival of 2–3 years.[1] Despite many promising preclinical studies and 125 phase II and phase III trials reported between 2008 and 2019, riluzole remains the only globally approved disease-modifying treatment, prolonging survival by an average of 2–3 months.[2] Edavarone, masitinib, AMX0035 (sodium phenylbutyrate and taursodeoxycholic acid) and tofersen have emerged as potentially promising candidates in clinical trials, but treatment effects are modest and none of these drugs have received approval in Europe.[3–7] In a long-term multi-centre prospective cohort study, edaravone showed no significant disease-modifying effect.[8] Previously, decisions to evaluate drugs in MND have been informed by preclinical studies, typically using mouse models, such as the $SOD1^{G93A}$ mouse, despite known limitations in the extent to which such models recapitulate human pathology,[9] and concerns of the reproducibility of findings from such models.[10] Clinical trials in MND are further complicated by the challenges of designing and delivering trials in a rapidly progressive, heterogeneous, disabling and fatal disease with a lack of reliable and sensitive outcome measures or biomarkers.[2]

Over the same period there have, however, been rapid technical advances in MND genomics, human induced pluripotent stem cells (iPSCs) and gene-editing, which have enabled better understanding of underlying pathophysiology (including potential shared pathways across neurodegenerative diseases), and the development of more sophisticated disease models. In parallel, drug repurposing (testing a drug already used or tested for other indications) has been successfully adopted in many diseases and can significantly reduce development time and cost, with the added benefit of the availability of prior safety data to guide selection.[11] In relapsing-remitting multiple sclerosis (MS), for instance, dimethyl fumarate, cladribine,[12] alemtuzumab[13 14] and rituximab[15] provide examples of successful repurposing as disease-modifying treatments.

Systematic review has been recommended to have a key role in planning new research studies.[16] We previously used a strategy based on systematic review to identify repurposed interventions for secondary progressive MS. This involved a two-stage systematic review and meta-analysis assessing clinical and preclinical data to identify putative therapeutic interventions[17] and led to the Multiple Sclerosis-Secondary Progressive Multi-Arm Randomisation Trial (MS-SMART), a phase IIb multi-arm randomised controlled trial.[18 19] The three drugs selected for MS-SMART were based in part on their availability for investigator-led clinical trials and did not show efficacy,

but two of the top seven drugs thus identified, ibudilast (ranked first), and lipoic acid, have since shown promise in phase II studies in secondary progressive MS.[20 21]

Noting similarities between MS and MND as neurodegenerative diseases with limited treatment options, in 2014 we embarked on a similar strategy to identify candidate oral neuroprotective agents in MND. In parallel, we developed the multi-arm multi-stage Motor Neuron Disease-Systematic Multi-Arm Adaptive Randomised Trial (MND-SMART, clinicaltrials.gov registration number: NCT04302870) to provide a more efficient pipeline to evaluate drugs in MND than conventional standalone two-arm trials.[22–24] Here, we describe the development and implementation of a systematic, structured and unbiased evidence-based approach to inform expert consensus in the selection of potential oral neuroprotective agents for clinical evaluation in MND-SMART. Specifically, the purpose here is not to provide a contemporary summary of existing evidence, but to describe the process through which clinical trial drugs were selected.

## METHODS

The work was guided by a systematic review protocol. Over the duration of the project and given the novelty of this approach, this protocol was updated in the light of accumulating experience, and the complete record of the protocol, including the changes made, is available at Open Science Framework.[25]

### Overview

The overall drug selection strategy is characterised in figure 1. We used systematic review to identify publications describing clinical trials or reports of the clinical use of drugs in MND and in four other neurodegenerative diseases which we considered might share pivotal pathways: Alzheimer's disease (AD), Parkinson's disease (PD), Huntington's disease (HD) and MS. For MS, we excluded studies of relapsing-remitting disease since we were interested in drugs tested in the progressive phase where neurodegeneration is a major feature. We also excluded studies of other diseases of motor neurons including Kennedy's disease and spinal muscle atrophy. We annotated publications for the drugs tested and diseases studied, taking forward drugs described in at least one MND publication or in publications in at least two other diseases. We scored each drug using a predefined framework evaluating efficacy, safety, study size and quality. In parallel, we performed a systematic review of the preclinical MND and frontotemporal dementia (FTD; because of pathological overlap with MND) literature for these drugs. We summarised evidence from both reviews for each drug and presented these to an expert panel consisting of clinical and academic neurologists with expertise in MND, clinical trials, pharmacology and preclinical models of MND.

### Systematic review of clinical evidence

The MS-SMART drug selection process used the same strategy, except we selected drugs tested at least once in

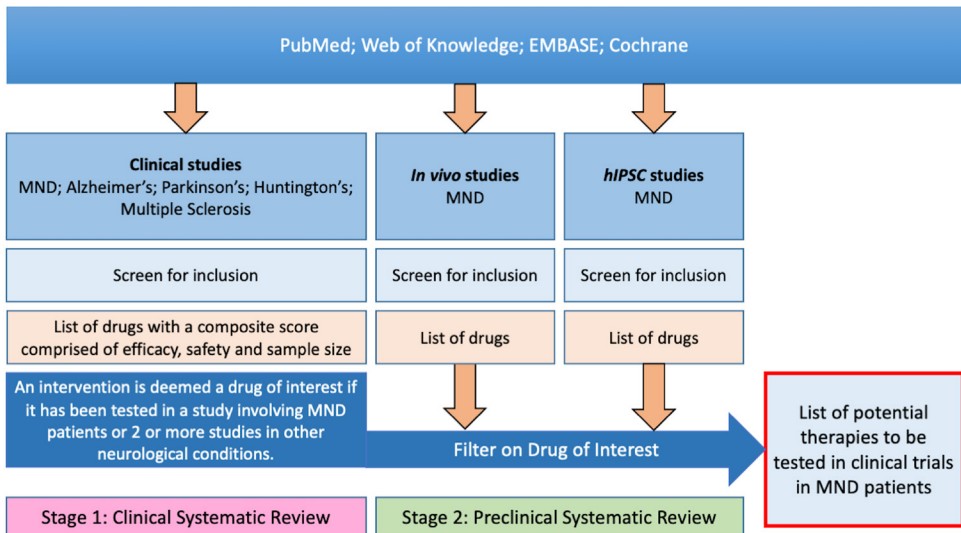

**Figure 1** Diagram illustrating two-stage systematic review approach to inform identification and selection of putative treatments to take forward to clinical trial. MND, motor neuron disease.

MS or in at least two other conditions. The protocol for and results from this search, conducted in September 2011, has been published.[17] That search involved three online databases (PubMed, ISI Web of Knowledge and EMBASE) using the terms "multiple sclerosis" OR "Alzheimer's disease" OR "Huntington's disease" OR "Parkinson's disease" OR "motor neuron disease" OR "amyotrophic lateral sclerosis". On 13 December 2013, we updated the search using the same terms but with limitations for PubMed to clinical trials, and date of record creation after 01/07/2011; for ISI Web of Knowledge to Document type 'Clinical trial' and publication years: 2011, 2012 and 2013; for EMBASE: previous search string AND ("case series" or "case report" or "cohort study"), with limits: human studies, full text studies from 2011; and we also contacted the Cochrane Neuromuscular review group to obtain a list of interventions tested in MND/ALS. The protocol of this update was stored locally; in the light of increasing recognition of the importance of making systematic review protocols available, the protocol was published without amendment in September 2019.[26]

Two reviewers (MM and KE) independently screened title and abstracts of publications identified in the new search against the inclusion and exclusion criteria (box 1) with discrepancies resolved by discussion. We included case reports, uncontrolled case series, non-randomised parallel group studies, crossover studies and randomised controlled trials with any report of safety or efficacy. We extracted basic information from each publication including author, year of publication, intervention tested and disease.

For all candidate interventions which had not been excluded based on feasibility or plausibility, we extracted further information on safety, efficacy, quality of study and study size from publications to a Microsoft Access database and scored these against a predefined metric (box 2 and table 1). For each drug, we calculated an overall drug score by taking the product of the mean score in each domain for safety, efficacy, quality, study size and multiplying this by $\log_{10}(1 + \text{number of publications})$. We then ranked drugs according to these scores.

### Systematic review of preclinical evidence
In parallel, we performed a systematic review of the preclinical literature (date of search 6 April 2016), focussing on publications describing the candidate interventions which had not been excluded on the basis of feasibility or plausibility, and using our previously published systematic review protocol.[27] We evaluated data from all in vivo models of MND and FTD including (1) mammalian models (mouse and rat), (2) organisms with a central nervous system (*Drosophila, Caenorhabditis elegans* and Zebrafish) and (3) multicellular eukaryotic models such as yeast. We also include data from studies using human iPSCs derived from people with MND.

### Patient and public involvement
The MND-SMART group has consulted people with MND, their families and carers via a patient and public involvement advisory group throughout the development of the trial. They expressed enthusiasm for a study design that enables definitive testing of drugs with promising efficacy, broad inclusion criteria and design features which minimise participant burden including remote study assessments, non-invasive outcome measures, liquid medication that can be administered in more advanced stages of disease, and drugs with favourable safety and tolerability profiles. This was taken into consideration by the expert panel during the drug selection process.

## Box 1 Eligibility criteria for clinical systematic review

Inclusion criteria
⇒ Publications reporting qualitative or quantitative data provided on either safety or efficacy of an orally delivered intervention in people with motor neuron disease (MND)/amyotrophic lateral sclerosis, Alzheimer's disease, Parkinson's disease, Huntington's disease or multiple sclerosis (MS).
⇒ Studies reporting change in clinical status (including death, tracheostomy free survival, relapse frequency, disability progression, behavioural symptoms), or changes in biomarkers (including MRI, blood, cerebrospinal fluid and muscle strength).

**Exclusion criteria**
⇒ Isolated reporting of non-pharmacological interventions such as acupuncture, aromatherapy, physiotherapy or exercise.
⇒ Articles reporting the use of interventions already licensed for clinical use in MND such as riluzole.
⇒ Articles on levodopa treatment for Parkinson's disease.
⇒ Studies reporting different modes of intervention delivery other than oral administration.
⇒ Publications reporting secondary analysis of previously published clinical trial data.
⇒ Protocols for clinical trials.
⇒ Preventative studies.
⇒ Reviews.
⇒ Studies on healthy volunteers.
⇒ Studies in patients with relapsing-remitting MS.
⇒ Studies reporting combination treatments including where an oral and a non-oral intervention are administered.
⇒ Publications where disease type is not specified to be in keeping with the included diseases (studies of vascular dementia, mild cognitive impairment and dementias other than Alzheimer's disease are excluded.
⇒ Studies on patients with parkinsonism are excluded as this do not imply Parkinson's disease exclusively).
⇒ Publications describing studies where multiple drugs were tested in a cohort without any data on individual drugs.

## Box 2 Scoring metric for clinical review

**Safety score (S)**
'Not described': 1 point.
'SUSARs (suspected unexpected serious adverse reactions) or mortality observed': 1 point.
'SAEs (serious adverse events) only': 2 points.
'AEs (adverse events) only': 3 points.
'No adverse effects reported': 4 points.
**Efficacy score (E)**
Efficacy score is assigned based on primary outcome measure, and where this is not identified, on the mean efficacy score for all outcomes reported in each publication.
'Not presented': 1 point.
'Definite (ie, statistically significant) worsening': 1 point.
'Neutral': 2 points.
'Non-significant improvement': 3 points.
'Significant improvement': 4 points.
**Quality score (Q)**
Study quality was assessed using a combination of criteria taken from a risk of bias tool developed through a Delphi process, GRADE and CAMARADES methods as shown in table 1. Once each publication has been scored they are sorted in quartiles of study quality based on the total number of checklist items scored, with the lowest quartile scoring 1one point and the highest quartile scoring 4 points.
**Study size score (SS)**
'1–10 participants': 1 point.
'11–100 participants': 2 points.
'101–1000 participants': 3 points.
'>1000 participants': 4 points.

### Shortlisting of drugs

Drugs with supportive evidence from both clinical and preclinical literature were shortlisted for review by an expert panel over two shortlisting rounds and a final selection round. Over the two shortlisting rounds, the panel rated drugs as 'green' (most favourable), 'amber' (less favourable) and 'red' (least favourable) based on biological plausibility; safety profile; and data from the clinical and preclinical reviews, and logistical considerations relating to factors including drug manufacturing, storage, dosing schedule and route of administration. Drugs rated 'red' for any criteria were excluded, along with drugs which had been tested in more than three previous trials in MND. Remaining drugs after the second shortlisting round entered a final selection round. We hand searched literature to identify and summarise all MND clinical trials for shortlisted drugs, including trials which may have been missed in the original search, trials which were annotated using drug synonyms in the original review (eg, acetylcysteine/acetylcystine/N-acetyl cystine/N-acetylcysteine for N-acetyl cysteine), and trials which have been excluded in the clinical review but

contain relevant data for expert panel discussions, such as drugs given in combination with other treatments or drugs given in non-oral formulations. We presented the expert panel with clinical, preclinical and clinical trial summaries for the final shortlisted drugs. Members of the expert panel independently ranked shortlisted drugs. The expert panel then met to finalise selection of drugs for clinical trial. As this approach might not cover novel drugs or pathways that had yet to be tested clinically in neurodegenerative diseases, the panel were given flexibility to consider emerging evidence for hitherto unconsidered drugs.

## RESULTS
### Clinical systematic review and initial screening of candidate interventions

The Preferred Reporting Items for Systematic Reviews and Meta-Analyses diagram for the clinical review is shown in figure 2. Further data are available in online supplemental file 1. Literature search in August 2011 of PubMed, ISI Web of knowledge and EMBASE, and Cochrane list of clinical trials in MS for MS-SMART identified 29 500 publications. Twelve thousand eight hundred and ninety-three duplicates were removed and 15 232 publications did not meet the inclusion criteria. One thousand three hundred and seventy-five publications were included in this initial search.

 Wong C, et al. BMJ Open 2023;13:e064169. doi:10.1136/bmjopen-2022-064169

**Table 1** Scoring method for evaluation of study quality in clinical systematic review

| | CAMARADES | Delphi | GRADE |
|---|---|---|---|
| **Binary response items** | | | |
| *Yes (1 point); no (0 points)* | | | |
| Peer reviewed publication | X | | |
| Statement of potential conflicts of interest | X | | |
| Sample size calculation | X | X | |
| Random allocation to group | X | X | X |
| Allocation concealment | X | | X |
| Blinded assessment of outcome | | X | |
| **Tertiary response items** | | | |
| *Yes (1 point); no (0 points); not clear (0.5 points)* | | | |
| Were the groups similar at baseline regarding the most important prognostic indicators? | | X | |
| Were the eligibility criteria specified? | | X | |
| Were point estimates and measures of variability presented for the primary outcome measures? | | X | |
| Was there intention to treat analysis? | | X | |
| Complete accounting of patient and outcome events | | | X |
| Non-selective outcome reporting | | X | |
| No other limitations | | | X |
| Can we be confident in the assessment of outcome? | | | X |
| **Quinary response items** | | | |
| *N/A; definitely yes (1 point); probably yes (0.75 points); probably no (0.25 points); definitely no (0 points)* | | | |
| Was selection of treatment and control groups drawn from the same population? | | | X |
| Can we be confident that patients received the allocated treatment? | | | X |
| Can we be confident that the outcome of interest was not present at start of the study? | | | X |
| Did the study stratify on variables associated with the outcome of interest or did the analysis take this into account? | | | X |
| Can we be confident in the assessment of the presence or absence of prognostic factors? | | | X |

Continued

**Table 1** Continued

| | CAMARADES | Delphi | GRADE |
|---|---|---|---|
| Was the follow-up of cohorts adequate? | | | X |
| Were cointerventions similar between groups? | | | X |

In the updated search in December 2013 a further 3124 publications were identified from PubMed, ISI Web of Knowledge, EMBASE and Cochrane databases. Five hundred and forty-one duplicates were removed, and 2322 publications did not meet the inclusion criteria. Two hundred and sixty-one publications were included.

Based on information contained in the title and abstract of these 1636 included publications we identified 595 interventions, of which 139 met our criteria of being described in at least one MND publication or in publications in two other diseases, in a total of 884 publications. On full text screening, 266 of these 884 publications did not meet our inclusion criteria. A further 50 interventions described in 90 publications were excluded because more detailed review of the primary literature at full text screening showed that the intervention has not been tested either in MND or in at least two of the other diseases. The remaining 66 interventions (528 publications) were scored against our predefined criteria and ranked (table 2). During preparation of this manuscript, we discovered that a publication describing the effect of N-acetyl cysteine in MND had been included in error, as no data were available for N-acetyl cysteine monotherapy.[28]

### Preclinical systematic review

We identified 14195 publications. After removing duplicates, two independent researchers screened title and abstract of 7586 unique publications, with differences reconciled by a third reviewer. 396 studies were included. Three hundred and thirty studies reported survival outcomes. Three hundred and thirteen studies reported behavioural outcomes. Of the 66 longlisted interventions from the clinical review, there were preclinical survival data for 20 drugs (table 3) and behavioural outcome data for 12 drugs.[29] Further data are available in online supplemental file 2.

### Shortlisted candidate drugs for clinical trial

Twenty-one drugs with supportive evidence in both clinical and preclinical systematic reviews were shortlisted for further evaluation. Simvastatin was added to the shortlist based on data from the clinical review and emerging data on its potential role in pathways of interest. Nuclear factor erythroid 2-related factor 2 (NRF2) is a transcription factor which controls expression and regulation of antioxidant proteins.[30] Modulating NRF2 may therefore protect against oxidative stress, a common feature across neurodegenerative diseases including MND.[30] A separate systematic review of interventions modulating NRF2

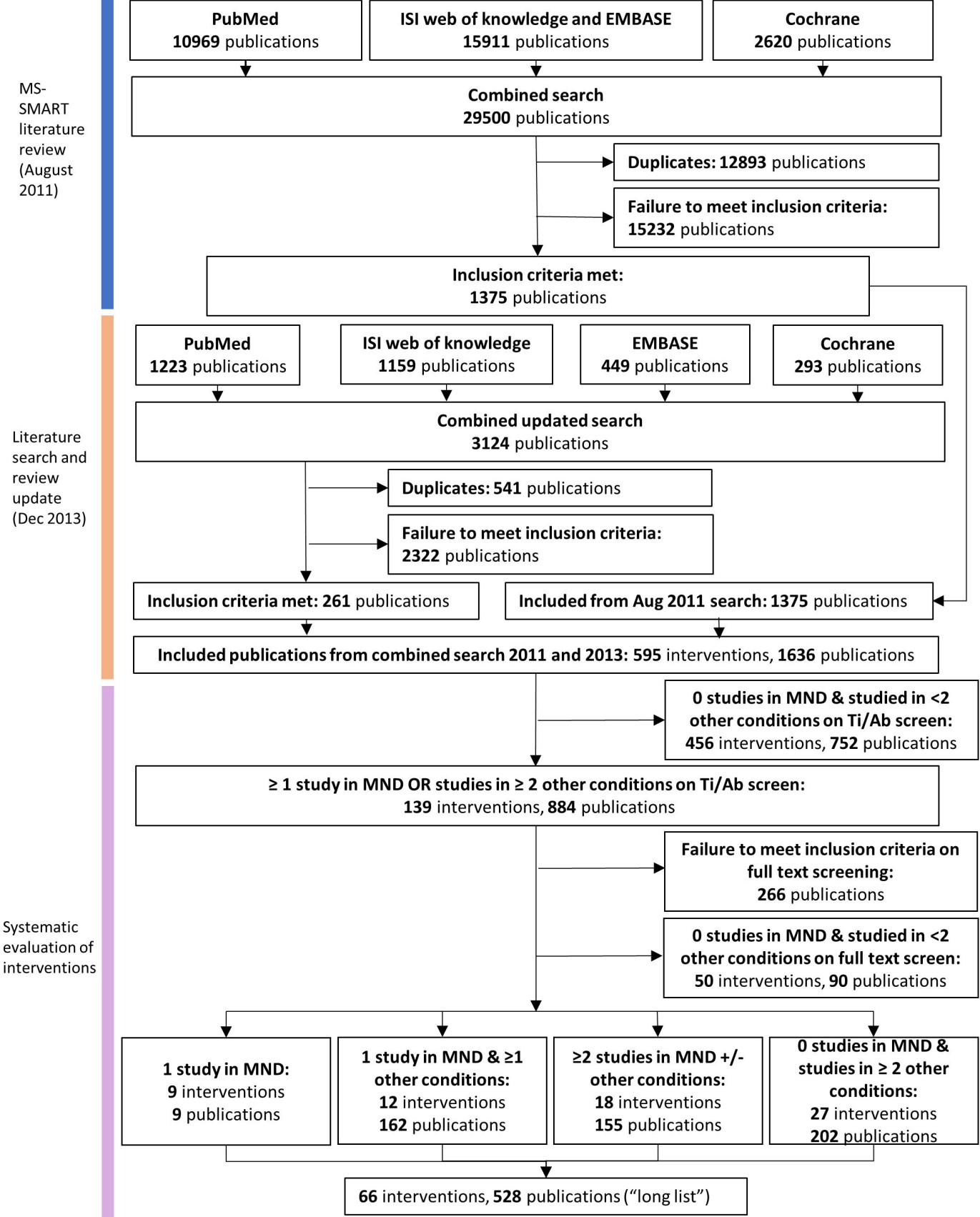

**Figure 2** Preferred Reporting Items for Systematic Reviews and Meta-Analyses diagram for clinical systematic review. MND, motor neuron disease; MS-SMART, Multiple Sclerosis-Secondary Progressive Multi-Arm Randomisation Trial; Ti/Ab, title/abstract.

**Table 2** Longlisted interventions ranked by drug score from clinical review

| Intervention | Number of publications | Quality score | Efficacy score | Safety score | Study size score | Drug score |
|---|---|---|---|---|---|---|
| Rivastigmine | 29 | 3.34 | 3.33 | 2.1 | 2.62 | 90.62 |
| Memantine | 51 | 3.02 | 2.87 | 2.2 | 2.47 | 80.7 |
| Vitamin D3 | 11 | 3.27 | 3.01 | 3.36 | 2.18 | 78.08 |
| Donepezil | 41 | 3.1 | 2.72 | 2.51 | 2.24 | 76.99 |
| Pramipexole | 14 | 3.07 | 3.04 | 2.43 | 2.64 | 70.41 |
| Galantamine | 12 | 2.75 | 2.85 | 2.67 | 2.5 | 58.29 |
| Amantadine | 59 | 2.37 | 3 | 2.36 | 1.9 | 56.53 |
| Dextromethorphan/quinidine | 3 | 4 | 3.33 | 2.33 | 3 | 56.19 |
| Selegiline | 21 | 3 | 2.59 | 2.1 | 2.38 | 51.98 |
| 4-aminopyridine | 10 | 3.2 | 2.76 | 2.5 | 2.1 | 48.28 |
| Acetyl-L-carnitine | 10 | 2.9 | 2.64 | 2.5 | 2.3 | 45.83 |
| Simvastatin | 5 | 3.6 | 2.27 | 2.8 | 2.4 | 42.73 |
| Lamotrigine | 6 | 4 | 2.25 | 2.33 | 2.33 | 41.41 |
| Bromocriptine | 13 | 2.92 | 2.4 | 2.54 | 1.92 | 39.21 |
| Clozapine | 6 | 3 | 2.9 | 2.5 | 2 | 36.8 |
| Gabapentin | 9 | 2.67 | 2.4 | 2.44 | 2.33 | 36.46 |
| Creatine | 12 | 2.67 | 2.14 | 2.33 | 2.42 | 35.87 |
| Ginkgo biloba | 3 | 3.67 | 2.61 | 3.67 | 1.67 | 35.21 |
| Minocycline | 11 | 2.45 | 2.27 | 2.64 | 2.18 | 34.54 |
| Vitamin E | 9 | 3.33 | 2.1 | 2 | 2.44 | 34.25 |
| Levetiracetam | 6 | 3 | 3.33 | 2.17 | 1.83 | 33.57 |
| Atomoxetine | 4 | 3.25 | 2.5 | 3.25 | 1.75 | 32.3 |
| Coenzyme Q10 | 9 | 3.33 | 2.16 | 1.89 | 2.33 | 31.67 |
| Tacrine | 10 | 3.3 | 2.81 | 1.8 | 1.8 | 31.26 |
| Olanzapine | 5 | 2.4 | 3.47 | 2.6 | 1.6 | 26.93 |
| Oestrogen | 9 | 3 | 2.66 | 1.67 | 2 | 26.57 |
| Nimodipine | 5 | 3 | 2.35 | 2.2 | 2.2 | 26.55 |
| Riluzole | 17 | 2.41 | 2.35 | 1.76 | 2.06 | 25.88 |
| Ciclosporin | 9 | 2.78 | 1.93 | 2.11 | 2.22 | 25.15 |
| Dextromethorphan | 7 | 2.29 | 2.4 | 2.71 | 1.86 | 24.97 |
| Naltrexone | 8 | 2.5 | 2.22 | 2.62 | 1.75 | 24.29 |
| Theophylline | 2 | 4 | 3.25 | 2.5 | 1.5 | 23.26 |
| Valproate | 9 | 2.56 | 2.33 | 2 | 1.89 | 22.53 |
| Fluoxetine | 6 | 2.67 | 2.42 | 2.33 | 1.67 | 21.18 |
| Levamisole | 3 | 2.67 | 2.78 | 2.33 | 2 | 20.81 |
| Melatonin | 8 | 2 | 2.11 | 2.75 | 1.88 | 20.81 |
| Celecoxib | 2 | 4 | 2 | 1.5 | 3 | 17.18 |
| 3,4-diaminopyridine | 6 | 2.83 | 2.4 | 2.17 | 1.33 | 16.62 |
| Milacemide | 4 | 3 | 1.75 | 2.25 | 2 | 16.51 |
| N-acetyl cystine | 1 | 3 | 3 | 3 | 2 | 16.26 |
| Tranylcypromine | 2 | 2.5 | 2.5 | 3.5 | 1.5 | 15.66 |
| Aspirin | 4 | 2.75 | 2 | 1.75 | 2.25 | 15.14 |
| Ursodeoxycholic acid | 1 | 4 | 2 | 3 | 2 | 14.45 |
| Tolbutamide | 2 | 3 | 2.5 | 2 | 2 | 14.31 |

Continued

| | | | | | | |
|---|---|---|---|---|---|---|
| **Table 2** Continued | | | | | | |
| **Intervention** | **Number of publications** | **Quality score** | **Efficacy score** | **Safety score** | **Study size score** | **Drug score** |
| Imipramine | 2 | 3.5 | 2 | 2 | 2 | 13.36 |
| Lithium | 12 | 2.42 | 2.19 | 1.5 | 1.5 | 13.29 |
| Modafinil | 2 | 4 | 3.33 | 1 | 2 | 12.72 |
| Omega 3 fatty acid | 2 | 2.5 | 1.75 | 3 | 2 | 12.52 |
| Octacosanol | 2 | 2.5 | 2 | 3.5 | 1.5 | 12.52 |
| Indinavir | 2 | 3.5 | 1.8 | 1.5 | 2.5 | 11.27 |
| Sodium phenylbutyrate | 1 | 4 | 2 | 2 | 2 | 9.63 |
| Tilorone | 1 | 4 | 2 | 2 | 2 | 9.63 |
| lipoic acid | 2 | 2.5 | 2 | 2 | 2 | 9.54 |
| Isoprinosine | 4 | 3 | 1.75 | 1.25 | 2 | 9.17 |
| Tetrahydrocannabinol | 2 | 3.5 | 1.5 | 1.5 | 2 | 7.51 |
| Topiramate | 1 | 4 | 1 | 2 | 3 | 7.22 |
| Haloperidol | 2 | 3 | 2.33 | 1 | 1.5 | 5.01 |
| Amino acid mixture | 5 | 1.6 | 2 | 1 | 2 | 4.98 |
| Rolipram | 2 | 1.5 | 2 | 3 | 1 | 4.29 |
| Alsamin | 1 | 2 | 3.5 | 1 | 2 | 4.21 |
| Pentoxifylline | 3 | 2 | 1.61 | 1 | 2 | 3.88 |
| Verapamil | 1 | 2 | 2 | 1 | 2 | 2.41 |
| IGF-1 | 1 | 2 | 1 | 1 | 3 | 1.81 |
| Propranolol | 1 | 1 | 3 | 1 | 2 | 1.81 |
| Fluvoxamine | 2 | 1 | 3 | 1 | 1 | 1.43 |
| Amitriptyline | 1 | 1 | 1 | 3 | 1 | 0.9 |

pathway in animal in vivo models of neurodegeneration and neuronal injury highlighted statins as a candidate drug targeting NRF2.[29 31] A clinical trial in AD demonstrated reduction of cerebrospinal fluid cholesterol level by simvastatin, thus demonstrating blood brain barrier penetrance, while providing supportive evidence that simvastatin may play a role in altering lipid biosynthesis, which may in turn inhibit protein misfolding and stress response mechanisms in MND.[32]

An evidence summary was compiled for each of the 22 shortlisted drugs including the following information: (1) if they had been tested in three or more in vivo MND studies, (2) the number of clinical trials in people with MND, (3) the putative target pathway, (4) feasibility for delivery via enteral tube (noting that swallowing is commonly affected in MND), (5) detailed safety information including common side effects, rare but serious side effects and requirements for monitoring, (6) published clinical studies in MND and (7) clinical trials registered on clinicaltrials.gov. The expert panel met on 9 January 2017 and discussed the evidence for each drug. Eleven drugs were excluded in the first round. Following a second round of discussions, four other drugs were excluded based on aggregate judgement of data presented. Reasons for exclusion are detailed in table 4.

The seven candidate drugs remaining were memantine, acetyl-l-carnitine, simvastatin, ciclosporin, melatonin, fluoxetine and N-acetyl cysteine. The clinical review data for each final shortlisted drug are summarised in table 5. MND clinical trials for shortlisted drugs including additional trials identified on handsearching are summarised in table 6.

On 30 January 2017, members of the expert panel independently ranked shortlisted drugs. On 2 February 2017, the panel reached a consensus to take the two top ranked drugs acetyl-l-carnitine and memantine forward to clinical trial. However, there were subsequent concerns regarding the availability of acetyl-l-carnitine without prescription and the resulting potential that self-medication with a known trial drug by trial participants, in addition to their randomised treatment allocation, might affect trial integrity.

Subsequently, the panel considered the other final shortlisted drugs and also considered emerging and compelling in vivo and in vitro evidence of the prevention of neurodegeneration by trazodone, through the targeting of eIF2a-P-mediated translational repression.[33] Following detailed consideration, the panel recommended memantine and trazodone as the first two investigational medicinal products for MND-SMART.

**Table 3** Summary of preclinical studies evaluating the effect of interventions longlisted from the clinical review on survival outcomes

| Publication | Drug | Total number of animals | Median survival in treatment group | Median survival in control group | LogMSR |
|---|---|---|---|---|---|
| Kira 2006 | Acetyl-L-carnitine | 20 | 270 | 240 | 0.1178 |
| Barneoud 1999 | Aspirin | 38 | 150 | 155 | −0.0328 |
| Tanaka 2011 | Bromocriptine | 69 | 40 | 35 | 0.1335 |
| Drachman 2002 | Celecoxib | 55 | 139 | 119 | 0.1554 |
| Karlsson 2004 | Ciclosporin | 13 | 144 | 130 | 0.1023 |
| Keep 2001 | Ciclosporin | 11 | 24 | 12 | 0.6931 |
| Turner 2003 | Clozapine | 16 | 140 | 132 | 0.0588 |
| Andreassen 2001 | Creatine | 24 | 155 | 135 | 0.1382 |
| Kaddurah-Daouk 2000 | Creatine | 13 | 169 | 144 | 0.1601 |
| Klivenyi 2004 | Creatine | 22 | 150 | 125 | 0.1823 |
| Choi 2008 | Oestrogen | 70 | 135 | 127 | 0.0611 |
| Koschnitzky 2014 | Fluoxetine | 34 | 139 | 132 | 0.0517 |
| Gurney 1996 | Gabapentin | 17 | 140 | 139 | 0.0072 |
| Gurney 1996 | Gabapentin | 38 | 175 | 165 | 0.0588 |
| Ferrante 2001 | Ginkgo biloba | 20 | 136 | 125 | 0.0843 |
| Fornai 2008 | Lithium | 20 | 146 | 117 | 0.2214 |
| Gill 2009 | Lithium | 55 | 124 | 127 | −0.0239 |
| Pizzasegola 2009 | Lithium | 20 | 119 | 129 | −0.0807 |
| Dardiotis 2013 | Melatonin | 28 | 143 | 143 | 0.0000 |
| Weishaupt 2006 | Melatonin | 50 | 137 | 131 | 0.0448 |
| Zhang 2013 | Melatonin | 30 | 145 | 137 | 0.0568 |
| Wang 2005 | Memantine | 21 | 130 | 122 | 0.0635 |
| Keller 2011 | Minocycline | 32 | 147 | 138 | 0.0632 |
| Kriz 2002 | Minocycline | 29 | 364 | 336 | 0.0800 |
| Van Den Bosch 2002 | Minocycline | 14 | 155 | 130 | 0.1759 |
| Zhang 2003 | Minocycline | 20 | 140 | 130 | 0.0741 |
| Zhu 2002 | Minocycline | 20 | 135 | 127 | 0.0611 |
| Andreassen 2000 | N-acetyl cysteine | 30 | 134 | 129 | 0.0380 |
| Jaarsma 1998 | N-acetyl cysteine | 28 | 251 | 239 | 0.0490 |
| Yip 2013 | Omega 3 | 32 | 182 | 182 | 0.0000 |
| Petri 2006 | Sodium phenylbutyrate | 26 | 139 | 127 | 0.0903 |
| Ryu 2005 | Sodium phenylbutyrate | 40 | 145 | 127 | 0.1325 |
| Crochemore 2009 | Valproate | 11 | 140 | 140 | 0.0000 |
| Rouaux 2007 | Valproate | 36 | 115 | 110 | 0.0445 |
| Sugai 2004 | Valproate | 17 | 295 | 265 | 0.1072 |
| Gianfocaro 2013 | Vitamin D | 100 | 126 | 124 | 0.0160 |

All listed studies used mouse models. LogMSR=log(median survival in treatment group/median survival in control group).

## DISCUSSION

Since drugs have undergone rigorous safety and pharmacokinetic testing, drug repurposing—the use of an established drug in a novel therapeutic indication—reduces costs and barriers to clinical development. Our experience of the successful application of a systematic approach to selecting neuroprotective drugs for repurposing in MS clinical trials[17] encouraged us to use a similar approach in MND. The first part of the review assessed clinical data in MND and in other neurodegenerative diseases with potential shared pathophysiological pathways. This allowed for the identification of drugs with good central nervous system penetrance and the potential for efficacy and safety in people with neurodegenerative diseases.

**Table 4** Drugs excluded following expert panel review and reasons for exclusion

| | Drug | Reason for exclusion |
|---|---|---|
| Excluded after round 1 | Bromocriptine | Unfavourable safety profile |
| | Gabapentin | >3 previous clinical trials in MND |
| | Creatine | >3 previous clinical trials in MND |
| | Clozapine | Unfavourable safety profile |
| | Minocycline | >3 previous clinical trials in MND |
| | Valproate | >3 previous clinical trials in MND |
| | Celecoxib | Unfavourable safety profile |
| | Aspirin | Poor biological plausibility |
| | Ginkgo biloba | Poor biological plausibility |
| | Lithium | >3 previous clinical trials in MND |
| | Amino acid mixture | >3 previous clinical trials in MND |
| Excluded after round 2 | Oestrogen | Aggregate judgement of data presented |
| | Vitamin D3 | |
| | Omega 3 | |
| | Sodium phenylbutyrate | |

MND, motor neuron disease.

However, drug selection based on clinical data alone is biased towards those tested in conditions where large well-designed randomised controlled trials have been performed and where the mechanism of action may be particular to that condition. Notably, two of our top five ranked drugs were cholinesterase inhibitors licensed for AD, a mechanism less relevant to MND. It was therefore important that we augment this approach with expert opinion and with preclinical data in MND and FTD models to provide mechanistic relevance. Taken together we have compiled evidence from clinical and preclinical data and used this to inform the selection of potential oral neuroprotective agents for clinical evaluation in people with MND. Through sequential systematic review, we identified a short list of 22 candidate interventions selected from an initial set of 595 drugs.

While some identified drugs demonstrate a good safety profile and have a relevant putative target pathway in MND, others have less favourable side effects profiles or a requirement for close therapeutic monitoring (eg, clozapine) which necessitates a higher threshold of evidence before testing in clinical trial. This highlights another advantage of our approach, in that it allows the identification of interventions that warrant further rigorous preclinical testing ('cislation'[34]) in vivo or in vitro models of ALS, with a view to providing more robust information for efficacy to support their inclusion in future clinical trials.

Following rounds of discussion, the expert panel identified memantine as a drug to be tested in MND-SMART. Memantine is a non-competitive N-methyl-D-aspartate receptor antagonist used in the treatment of moderate to severe AD. It was shown to significantly delay disease progression and improve survival in mouse models carrying a high copy number of $SOD1^{G93A}$.[35] Memantine has been previously tested in three MND clinical trials. A phase II double-blind placebo-controlled study of 63 participants with ALS powered to evaluate safety and tolerability did not identify any increase in adverse events.[36] There was a trend towards improvement in participants treated with memantine 20 mg/day, but no significant difference in Amyotrophic Lateral Sclerosis Functional Rating Scale. In a 5-month randomised double-blind study of 24 participants with ALS, there was a significant slowing of spinal motor neuron loss as demonstrated on motor unit estimation testing in the high dose group (10 mg two times a day) compared with low dose (5 mg two times a day).[37] Adverse events were not reported. In a single-arm pilot study of 19 participants with ALS, participants treated with riluzole and memantine had reduction in rate of Amyotrophic Lateral Sclerosis Functional Rating Scale decline and reduced cerebrospinal fluid (CSF) tau levels without any increase in adverse events.[38]

We also asked the expert panel to consider other drugs for which relevant data had only become available after the searches described here had been performed. Trazodone was nominated for consideration through

**Table 5** Clinical systematic review data for the final shortlisted drugs: number of publications (including interventional and observational studies) and participants according to type of disease

| Drug | Number of publications | | | | | | Number of participants | | | | | |
|---|---|---|---|---|---|---|---|---|---|---|---|---|
| | MND | AD | HD | MS | PD | Total | MND | AD | HD | MS | PD | Total |
| Acetyl-L-carnitine | 0 | 9 | 1 | 0 | 0 | **10** | 0 | 1224 | 10 | 0 | 0 | **1234** |
| Ciclosporin | 2 | 0 | 0 | 7 | 0 | **9** | 110 | 0 | 0 | 1092 | 0 | **1202** |
| Fluoxetine | 0 | 0 | 1 | 2 | 3 | **6** | 0 | 0 | 30 | 51 | 32 | **113** |
| Melatonin | 1 | 4 | 0 | 0 | 3 | **8** | 3 | 273 | 0 | 0 | 64 | **340** |
| Memantine | 1 | 32 | 2 | 1 | 15 | **51** | 63 | 11912 | 39 | 116 | 809 | **12939** |
| N-acetyl cysteine | 0 | 1 | 0 | 0 | 0 | **1** | 0 | 47 | 0 | 0 | 0 | **47** |
| Simvastatin | 0 | 3 | 0 | 1 | 1 | **5** | 0 | 469 | 0 | 307 | 12 | **788** |

AD, Alzheimer's disease; HD, Huntington's disease; MND, motor neuron disease; MS, multiple sclerosis; PD, Parkinson's disease.

**Table 6** Summary of previous motor neuron disease clinical trials for final shortlisted drugs

| Drug | Publication | RCT | Number of participants (active arm: placebo arm) | Duration | Primary outcome measure | Efficacy results | Safety results |
|---|---|---|---|---|---|---|---|
| Memantine | De Carvalho 2010 | Y | 63 (32:31) | 12 months | ALSFRS and safety | Equivocal (underpowered) | No increase in AEs |
| | Levine 2010 (with riluzole) | N | 20 (20:0) | 18 months | Safety and tolerability | ALSFRS decline of −0.73 points/month (pretreatment rate −1.07/month) | AE: nausea in one participant |
| | Chan 2011 | N | 24 (24:0) | 5 months | MUNE, MRSI, ALSFRS-R and MMT | MUNE: significant slowing of MN loss (−12.4±3.7 / month in run-in phase to −5.3±2.2/month in treatment phase; mean±SD, p=0.03). Other outcomes equivocal | AE: similar between run-in and treatment phases |
| Acetyl-L-carnitine | Beghi 2013 (with riluzole) | Y | 82 (42:40) | 12 months | Proportion of participants no longer self-sufficient | Significantly less treated participants loss self-sufficiency (80.9% ALC vs 97.5% placebo, p=0.0296) | No significant difference in AEs |
| Simvastatin | Nil | | | | | | |
| Ciclosporin | Appel 1988 | Y | 74 (36:38) | 48 weeks | Appel ALS rating scale | Progression to 22 points equivocal in treated and untreated participants (relative risk of ciclosporin 0.991, p=0.485) In subgroup of male participants with symptoms ≤18 months, relative risk of progression was 0.302, p=0.0205 | Significant number of expected ARs: hirsutism, headache, flushing, nausea and vomiting, tremor, anorexia and gum hyperplasia. No SUSARs |
| Melatonin | Weishaupt 2006 (rectal melatonin) | N | 31 (31:0) | 12 months | Safety | ALSFRS presented without any quantitative analysis | No AEs reported or observed |
| Fluoxetine | Nil | | | | | | |
| N-acetyl cysteine | Louwerse 1995 (acetylcysteine) | Y | 110 (54:56) | 12 months | Survival | Non-significant trend towards improvement in survival (HR 0.74 in acetylcysteine group compared with placebo, 95% CI 0.41 to 1.33; log-rank test, p=0.31) | No safety data reported |

AEs, adverse events; ALC, acetyl-L-carnitine; ALSFRS, Amyotrophic Lateral Sclerosis Functional Rating Scale; ALSFRS-R, Amyotrophic Lateral Sclerosis Functional Rating Scale-Revised; ARs, adverse reactions; 95% CI, 95% confidence interval; HR, Hazard ratio; MMT, manual muscle testing; MN, motor neuron; MND, motor neuron disease; MRSI, magnetic resonance spectroscopy imaging; MUNE, motor unit number estimate; RCT, randomised controlled trial; SUSAR, suspected unexpected severe adverse reaction.

this route. Trazodone is an atypical serotonin antagonist and reuptake inhibitor antidepressant. An unbiased drug screen found that trazodone inhibited Protein Kinase RNA-like endoplasmic reticulum kinase (PERK), which is pivotal to stress granule formation, a common feature of neurodegenerative diseases.[33] Inhibition of PERK was found to be beneficial in a fly model of ALS as well as in an in vitro neuronal assay of TDP-43 injury.[39] Furthermore, trazodone has been shown to modulate the ER-stress response resulting in an improvement in survival in animal models of prion disease and FTD.[33] Trazodone also modulated mitochondrial energy metabolism and fatty acid synthesis in animal models of HD, and may prevent mitochondrial dysfunction in MND.[40] In a randomised double-blind placebo-controlled crossover phase II trial in 31 participants with FTD, trazodone was found to improve cognition as assessed by the neuropsychiatric inventory.[41] In trials of trazodone in PD and AD, although there was no improvement in cognition, symptoms of sleep disturbance and depression were alleviated and adverse events were not increased.[42 43]

### Limitations of this approach

The main challenge in this approach to drug selection is the ambition to base drug choice on the most contemporary evidence. Systematic reviews are time consuming, as evidenced by the interval between our updated search (2013) and expert committee consideration (2017). Furthermore, drugs with promising data in some domains would be excluded if they have been tested in only one disease other than MND; or if they have not been tested clinically despite overwhelming preclinical evidence. We excluded combination therapies, but it may be—as in the treatment of various cancers[44] and infections[45 46]—that engagement with multiple targets is required to achieve a substantial disease-modifying effect.

The inclusion of trazodone may be seen as a weakness of this approach, but in our view this demonstrates a strength in the flexibility of our approach. We do not believe that systematic review should be used as part of a rigid selection process with little need for input from experts; but rather that expert input is informed by a detailed and robust systematic review process. The expert committee selected trazodone in the full knowledge that it had not been selected through the systematic review process, but were convinced that the emerging evidence of potential efficacy, coupled with long standing clinical experience in its use, made it an attractive candidate for testing in MND-SMART.

Finally, some have suggested that the literature-based systematic review approach to drug selection is intrinsically flawed because it does not take into account disease specific pathophysiology (which may be largely unknown).[47] While the three drugs tested in MS-SMART were not effective,[18] we note that two other drugs on the final MS-SMART shortlist - ibudilast[20] and lipoic acid[21]— have since shown promise in independent phase II trials. Lipoic acid has been identified again as a favourable candidate drug in a further, independent review in 2020.[48] We sought to address this issue here by considering, in addition to clinical information, data from in vivo and in vitro research. Although much successful drug repurposing has been opportunistic and serendipitous, we recognise that future efforts should include consideration of our mechanistic understanding of neurodegenerative diseases and should systemically incorporate additional target and pathway-based information.[11]

### Future approaches to drug selection in MND-SMART

Ongoing rounds of drug selection for MND-SMART exploit innovations in automating literature searches, screening and annotation, with these algorithms trained using the human efforts in the work reported here. These techniques show substantial improvements in efficiency in other fields.[49] Using the Systematic Review Facility (SyRF) (https://syrf.org.uk)[50] we have enabled a 'living' systematic review with automatic search, citation screening, identification of disease and drug, and selection of drugs meeting our criteria for the range of diseases in which studies have been performed. Because of similarities between MND and FTD we have included this as an additional disease of interest. Further details are extracted from full text publications of shortlisted drugs by a combination of machine and human work enabled through the SyRF platform, with human monitoring of machine decisions. The incorporation of machine learning and text mining techniques substantially reduces the human effort required and makes this approach feasible in the context of timely drug selection for adaptive clinical trials.

Complementing our literature-based approach, our current platform incorporates data from additional domains, including in house in vitro high throughput screening using human induced pluripotent stem cell culture; pathway and network analysis; and mining of drug and trial databases. We have also sought a broader range of inputs to our expert committee such that it now includes those with experience and expertise in managing people with MND and their symptoms, and of clinical trials, translational and clinical neurology, systematic reviews, experimental drug screening, pharmacology, chemistry, and drug discovery.

### CONCLUSIONS

We describe our experience in conducting a systematic, structured, unbiased and evidence-based approach to the selection of candidate drugs for evaluation in a clinical trial in MND by combining review of clinical and preclinical literature, and expert panel input. The first two drugs selected are memantine and trazodone. For future selection, we will incorporate machine learning and text mining to our systematic reviews and data from our drug discovery platform.

**Author affiliations**
[1]Anne Rowling Regenerative Neurology Clinic, The University of Edinburgh, Edinburgh, UK

<sup>2</sup>Euan MacDonald Centre for Motor Neuron Disease Research, The University of Edinburgh, Edinburgh, UK
<sup>3</sup>Centre for Clinical Brain Sciences, The University of Edinburgh, Edinburgh, UK
<sup>4</sup>Medical Research Council Clinical Trials Unit at UCL, Institute of Clinical Trials and Methodology, University College London, London, UK
<sup>5</sup>Institute of Medical Sciences, University of Aberdeen, Aberdeen, UK
<sup>6</sup>Computer and Information Science, University of Strathclyde, Glasgow, UK
<sup>7</sup>Edinburgh Medical School, The University of Edinburgh, Edinburgh, UK
<sup>8</sup>Royal Infirmary of Edinburgh, NHS Lothian, Edinburgh, UK
<sup>9</sup>UK Dementia Research Institute, University of Edinburgh, Edinburgh, UK
<sup>10</sup>Borders General Hospital, NHS Borders, Melrose, UK
<sup>11</sup>College of Medicine and Veterinary Medicine, The University of Edinburgh, Edinburgh, UK
<sup>12</sup>Institute of Neurological Sciences, NHS Greater Glasgow and Clyde, Glasgow, UK
<sup>13</sup>School of Psychology and Neuroscience, University of Glasgow, Glasgow, UK
<sup>14</sup>Centre for Discovery Brain Sciences, The University of Edinburgh, Edinburgh, UK
<sup>15</sup>Neurology Department, NHS Forth Valley, Stirling, UK
<sup>16</sup>Institute of Evolutionary Biology, The University of Edinburgh, Edinburgh, UK
<sup>17</sup>Queen Square Multiple Sclerosis Centre, Department of Neuroinflammation, UCL Queen Square Institute of Neurology, London, UK
<sup>18</sup>University College London Hospitals, Biomedical Research Centre, National Institute for Health Research, London, UK

**Twitter** Charis Wong @DrCharisWong, Jenna M Gregory @jennagregory488, Maarij Anwar @Maarij_Anwar, Victoria Collins @VGCollins__, Peter Foley @peterfoley10, Stella A Glasmacher @StellaGlasmach1, Gavin Langlands @GavinLanglands, D Leighton @Yelleighton, Arpan R Mehta @DrArpan100, Ankur Singh @AnkurP_Singh, Fergal M Waldron @FergalWaldron, Bhuvaneish T Selvaraj @bhuvaneish, Suvankar Pal @suvankarpal and Malcolm Macleod @Maclomaclee

**Contributors** SP, SC and MM managed the project. CW, JMG, JLiao, KE, HMV and MM were project administrators. JMG, KE, HMV, SC and MM conceptualised the project. CW, JMG, JLiao, KE, HMV, AAK, MA, CB, FSB, JC, AC, JYC, CC, VC, JD, EE, PF, YCF, LF-H, ABG, SAG, ÁH, KJ, NJ, AK, JK, GL, DL, JLiu, JLyon, ARM, AM, VN, NHP, SQ, YR, ASalzinger, BS, ASingh, TS, AT, OT, FMW, SP and MM performed systematic searching, screening, annotation and data extraction. CW, JMG, JLiao, KE, HMV and MM developed the project methodology, curated and analysed data. JLiao handled software and programming. CW developed data visualisation. AC and BTS performed in vitro drug screening. JC, RS, PC, SP, SC and MM were members of the expert panel. CW and JMG wrote the original draft. CW, JMG, JLiao, JC, ARM, JC, RS, SP, SC and MM reviewed and edited the manuscript. All authors have read and approved the manuscript. MM is the guarantor of the overall content: The guarantor accepts full responsibility for the finished work and/or the conduct of the study, had access to the data, and controlled the decision to publish.

**Funding** For the purpose of open access, the authors have applied a Creative Commons Attribution (CC BY) licence to any Author Accepted Manuscript version arising from this submission. MND-SMART is funded by grants from MND Scotland, My Name'5 Doddie Foundation (DOD/14/15) and specific donations to the Euan MacDonald Centre. The Chandran lab is supported by the UK Dementia Research Institute, which receives its funding from UK DRI Ltd, funded by the UK Medical Research Council, Alzheimer's Society and Alzheimer's Research UK. EE is a clinical academic fellow jointly funded by MND Scotland (MNDS) and the Chief Scientist Office (CSO) (217ARF R45951). ARM was a Lady Edith Wolfson Clinical Fellow, jointly funded by the Medical Research Council (MRC) and the Motor Neurone Disease Association (MR/R001162/1). ASalzinger is funded by Marie Sklodowska-Curie actions Innovative Training Network (ITN). BTS is funded by Rowling fellowship.

**Competing interests** In the last 3 years, JC has received support from the Efficacy and Evaluation (EME) Programme, a Medical Research Council (MRC) and National Institute for Health Research (NIHR) partnership and the Health Technology Assessment (HTA) Programme (NIHR), the UK MS Society, the US National MS Society and the Rosetrees Trust. He is supported in part by the NIHR University College London Hospitals (UCLH) Biomedical Research Centre, London, UK. He has been a local principal investigator for a trial in MS funded by the Canadian MS society. A local principal investigator for commercial trials funded by: Actelion, Novartis and Roche; and has taken part in advisory boards/consultancy for Azadyne, Janssen, Merck, NervGen, Novartis and Roche.

**Patient and public involvement** Patients and/or the public were involved in the design, or conduct, or reporting, or dissemination plans of this research. Refer to the Methods section for further details.

**Patient consent for publication** Not applicable.

**Ethics approval** Not applicable.

**Provenance and peer review** Not commissioned; externally peer reviewed.

**Data availability statement** All data relevant to the study are included in the article or uploaded as supplementary information.

**ORCID iDs**
Charis Wong http://orcid.org/0000-0002-8488-037X
Kieren Egan http://orcid.org/0000-0002-1639-4281
Maarij Anwar http://orcid.org/0000-0002-7136-8342
Victoria Collins http://orcid.org/0000-0003-0561-096X
Stella A Glasmacher http://orcid.org/0000-0003-1165-9153
Áine Heffernan http://orcid.org/0000-0002-4991-3949
Olaf Tomala http://orcid.org/0000-0002-9757-6932
Jeremy Chataway http://orcid.org/0000-0001-7286-6901
Suvankar Pal http://orcid.org/0000-0003-4276-639X
Siddharthan Chandran http://orcid.org/0000-0001-6827-1593
Malcolm Macleod http://orcid.org/0000-0001-9187-9839

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
