## [Reviewer comments · BMJ Open]

ARTICLE DETAILS

TITLE (PROVISIONAL)	A systematic, comprehensive, evidence-based approach to identify neuroprotective interventions for motor neuron disease: using systematic reviews to inform expert consensus
AUTHORS	Wong, Charis; Gregory, Jenna M.; Liao, Jing; Egan, Kieren; Vesterinen, Hanna M.; Ahmad Khan, Aimal; Anwar, Maarij; Beagan, Caitlin; Brown, Fraser; Cafferkey, John; Cardinali, Alessandra; Chiam, Jane Yi; Chiang, Claire; Collins, Victoria; Dormido, Joyce; Elliott, Elizabeth; Foley, Peter; Foo, Yu Cheng; Fulton-Humble, Lily; Gane, Angus; Glasmacher, Stella; Heffernan, Áine; Jayaprakash, Kiran; Jayasuriya, Nimesh; Kaddouri, Amina; Kiernan, Jamie; Langlands, Gavin; Leighton, D; Liu, Jiaming; Lyon, James; Mehta, Arpan; Meng, Alyssa; Nguyen, Vivienne; Park, Na Hyun; Quigley, Suzanne; Rashid, Yousuf; Salzinger, Andrea; Shiell, Bethany; Singh, Ankur; Soane, Tim; Thompson, Alexandra; Tomala, Olaf; Waldron, Fergal; Selvaraj, Bhuvaneish; Chataway, Jeremy; Swingler, Robert; Connick, Peter; Pal, Suvankar; Chandran, Siddharthan; Macleod, Malcolm

VERSION 1 – REVIEW

REVIEWER	Pennuto, Maria University of Padua, Biomedical Sciences
REVIEW RETURNED	24-Jun-2022

GENERAL COMMENTS	in this metanalysis, the authors review previous work on clinical trials and preclinical assessment of the efficacy of specific therapeutic approaches. They include neurodegenerative diseases, such as AD and PD. However, they refer to MNDs, which is broad, while they focus on ALS, without including SBMA (Kennedy disease), another neurodegenerative disease. They may also consider including SMA, if there is anything that is overlapping, like IGF1 trials.
--

REVIEWER	Cunniffe, Nick Cambridge University, Department of clinical neurosciences
REVIEW RETURNED	11-Jul-2022

GENERAL COMMENTS	Thank you for asking me to review this important, timely, and high quality manuscript. This research describes the process undertaken to enable a robust, evidence-based and unbiased approach to selecting drugs for repurposing in the MND-SMART clinical trial. Given the unmet need of neuroprotective strategies in this condition, and the myriad benefits to be had from drug repurposing, this is a very important piece of work for people working to find effective treatments for MND. But the method (and discussion of its benefits and limitations)
--

	is also highly valuable for researchers in other disease areas. The authors present their two-stage procedure to identify drugs with clinical and preclinical rationale, ultimately settling on memantine and trazodone as their immediate priorities following panel review. The work undertaken here is considerable and I think highly effective at distilling information from multiple sources to allow effective decision making. The authors have additionally reflected on previous likeminded efforts to select drugs for repurposing in other illnesses such as multiple sclerosis. My minor comments are as follows: Despite the enormous body of work in the clinical and preclinical stage, only one of the final two drugs taken forward by the panel came from this process. The reasons for trazodone's addition are clearly presented in the discussion. But I would be very interested to learn about the decision making process by the panel: did they score/rank the final 7 alongside trazodone? Was there consensus between the group (and if not, how was this resolved)? And what were the negatives about the other 6 drugs to make trazodone trump each of these? A strength of the study was to synthesise the thorough method of evaluating each intervention with the views of those affected by MND. I wonder if comment could be added to the results or discussion as to how this impacted the choice of drugs. At the moment it reads as if the only thing contributed was to consider liquid formulations (and I suspect a lot more value was added by their participation than this). I believe readers from other disease areas would be interested to learn more about how PPI has been integrated in this method. I can not rationalise the numbers in Table 7 and 8 when taken together. For instance, the number of MND publications for memantine in table 7 is given as 1, and in table 8 it then cites 3 MND clinical trials with 63+20+24 participants (rather than the 63 number in table 7). Similarly, ciclosporin has 0 clinical studies in MND according to table 7, but then cites one of 74 participants in table 8? Are the numbers in table 7 accurate, or is it perhaps the titles/legends of table 7/8 that need clarification? Under strengths and limitations. Should these be in the present tense given this is typically at the start of the paper: 'we describe', 'provide' etc.? Page 13 Line 47 - 'described in 90 publications were were excluded because'. Were is duplicated here In the results section 'shortlisted candidate drugs for clinical trial', I wonder if the dates are accurate, as my read of this is that they met in 2017 to shortlist 22 drugs, and then a detailed summary was compiled for these, before then met (?again) on 9th January 2017. Did this process really take place within a week? Are the dates accurate? Could this be reworded just a touch to make it clearer the stages that were taken by the panel?
--	---

VERSION 1 – AUTHOR RESPONSE

Reviewer 1	
In this metanalysis, the authors review previous work on clinical trials and preclinical assessment of the efficacy of specific therapeutic approaches. They include neurodegenerative diseases, such as AD and PD. However, they refer to MNDs, which is broad, while they focus on ALS, without including SBMA (Kennedy disease), another neurodegenerative disease. They may also consider including SMA, if there is anything that is overlapping, like IGF1 trials.	Thank you. To clarify, we use the term 'motor neuron disease' here in line with common use in the United Kingdom (such as by the Motor Neuron Disease Association, Motor Neuron Disease Scotland and in the MND-SMART trial) to refer to amyotrophic lateral sclerosis and associated conditions (primary lateral sclerosis and progressive muscular atrophy), rather than all diseases of the motor neurons. We have clarified this in the text. For the purpose of identifying candidate drugs for this population, we were keen to include evidence from other neurodegenerative conditions which may share similar pathways and from similar population demographics. We decided to include data from AD, PD, HD and progressive MS as these are the most common neurodegenerative diseases in adulthood. We will consider including other diseases of motor neuron including SBMA and SMA in future evidence synthesis.
Reviewer 2	
Despite the enormous body of work in the clinical and preclinical stage, only one of the final two drugs taken forward by the panel came from this process. The reasons for trazodone's addition are clearly presented in the discussion. But I would be very interested to learn about the decision making process by the panel: did they score/rank the final 7 alongside trazodone? Was there consensus between the group (and if not, how was this resolved)? And what were the negatives about the other 6 drugs to make trazodone trump each of these?	Thank you. We have now provided additional context in the results and discussion section on the selection of trazodone.
A strength of the study was to synthesise the thorough method of evaluating each intervention with the views of those affected by MND. I wonder if comment could be added to the results or discussion as to how this impacted the choice of drugs. At the moment it reads as if the only thing contributed was to consider liquid formulations (and I suspect a lot more value was added by their participation than this). I believe readers from other disease areas would be interested to learn more about how PPI has been integrated in this method	Thank you for your comment. For the initial arms of MND-SMART, our PPI advisory group expressed enthusiasm for a study design that enables definitive testing of drugs with promising efficacy, broad inclusion criteria, and design features which minimise participant burden including remote study assessments, non-invasive outcome measures, liquid medication that can be administered in more advanced stages of disease, and drugs with favourable safety and tolerability profiles. We have now added this to the manuscript. We are working towards having more patient and public involvement in drug selection for future arms.
I can not rationalise the numbers in Table 7 and 8 when taken together. For instance, the number of MND publications for memantine in table 7 is given as 1, and in table 8 it then cites 3 MND clinical trials with 63+20+24 participants	Thank you for flagging this up. We have added some clarification around additional evidence summaries for the final shortlisted drugs, as follows, and included

(rather than the 63 number in table 7). Similarly, ciclosporin has 0 clinical studies in MND according to table 7, but then cites one of 74 participants in table 8? Are the numbers in table 7 accurate, or is it perhaps the titles/legends of table 7/8 that need clarification?	clarification in the caption for Table 7 (that this is from the clinical systematic review and includes interventional and observational data), and in Table 8, where we have added notes for additional trials which were identified subsequently. “We handsearched literature to identify and summarise all MND clinical trials for shortlisted drugs (Table 8), including trials which may have been missed in the original search, trials which were annotated using drug synonyms in the original review (e.g. acetylcysteine/acetylcystine/N-acetyl cystine/N-acetylcysteine for N-acetyl cysteine), and trials which have been excluded in the clinical review but contain relevant data for expert panel discussions, such as drugs given in combination with other treatments or drugs given in non-oral formulations.”
Under strengths and limitations. Should these be in the present tense given this is typically at the start of the paper: ‘we describe’, ‘provide’ etc.?	Thank you for this feedback. We have now changed this to present tense in line with the rest of the text.
Page 13 Line 47 - ‘described in 90 publications were were excluded because’. Were is duplicated here	Yes, thank you for flagging this up – now corrected.
In the results section ‘shortlisted candidate drugs for clinical trial’, I wonder if the dates are accurate, as my read of this is that they met in 2017 to shortlist 22 drugs, and then a detailed summary was compiled for these, before then met (?again) on 9th January 2017. Did this process really take place within a week? Are the dates accurate? Could this be reworded just a touch to make it clearer the stages that were taken by the panel?	We have now provided additional details of the shortlisting process.

VERSION 2 – REVIEW

REVIEWER	Cunniffe, Nick Cambridge University, Department of clinical neurosciences
REVIEW RETURNED	20-Dec-2022
GENERAL COMMENTS	Thank you for asking me to review this revised manuscript. It is significantly improved and I feel my comments have been addressed. I have no further recommendations